# Relationships between Osteopontin, Osteoprotegerin, and Other Extracellular Matrix Proteins in Calcifying Arteries

**DOI:** 10.3390/biomedicines12040847

**Published:** 2024-04-11

**Authors:** Aleksandra Kuzan, Agnieszka Chwiłkowska, Krzysztof Maksymowicz, Urszula Abramczyk, Andrzej Gamian

**Affiliations:** 1Department of Preclinical Sciences, Pharmacology and Medical Diagnostics, Faculty of Medicine, Wrocław University of Science and Technology, Wybrzeże Stanisława Wyspiańskiego 27, 50-370 Wroclaw, Poland; aleksandra.kuzan@pwr.edu.pl; 2Department of Molecular and Cellular Biology, Faculty of Pharmacy, Wroclaw Medical University, Borowska 211A, 50-556 Wroclaw, Poland; agnieszka.chwilkowska@umw.edu.pl; 3Department of Forensic Medicine, Faculty of Medicine, Wroclaw Medical University, J. Mikulicza-Radeckiego 4, 50-345 Wroclaw, Poland; krzysztof.maksymowicz@umw.edu.pl; 4Department of Pediatric Cardiology, Regional Specialist Hospital, Research and Development Center, Kamieńskiego 73A, 51-124 Wroclaw, Poland; 5Department of Immunology of Infectious Diseases, Hirszfeld Institute of Immunology and Experimental Therapy, Polish Academy of Sciences, Rudolfa Weigla 12, 53-114 Wroclaw, Poland; andrzej.gamian@hirszfeld.pl

**Keywords:** calcification, atherosclerosis, osteoprotegerin, osteopontin, extracellular matrix proteins

## Abstract

Osteopontin (OPN) and osteoprotegerin (OPG) are glycoproteins that participate in the regulation of tissue biomineralization. The aim of the project is to verify the hypothesis that the content of OPN and OPG in the aorta walls increases with the development of atherosclerosis and that these proteins are quantitatively related to the main proteins in the extracellular arteries matrix. Quantitative and qualitative analyses of the OPN and OPG content in 101 aorta sections have been conducted. Additionally, an enzyme-linked immunosorbent assay (ELISA) test has been performed to determine the collagen types I–IV and elastin content in the tissues. Correlations between the biochemical data and patients’ age/sex, atherosclerosis stages, and calcification occurrences in the tissue have been established. We are the first to report correlations between OPN or OPG and various types of collagen and elastin content (OPG/type I collagen correlation: r = 0.37, *p* = 0.004; OPG/type II collagen: r = 0.34, *p* = 0.007; OPG/type III collagen: r = 0.39, *p* = 0.002, OPG/type IV collagen: r = 0.27, *p* = 0.03; OPG/elastin: r = 0.42, *p* = 0.001; OPN/collagen type I: r = 0.34, *p* = 0.007; OPN/collagen type II: r = 0.52, *p* = 0.000; OPN/elastin: r = 0.61, *p* = 0.001). OPN overexpression accompanies calcium deposit (CA) formation with the protein localized in the calcium deposit, whereas OPG is located outside the CA. Although OPN and OPG seem to play a similar function (inhibiting calcification), these glycoproteins have different tissue localizations and independent expression regulation. The independent expression regulation presumably depends on the factors responsible for stimulating the synthesis of collagens and elastin.

## 1. Introduction

Cardiovascular diseases have been the leading cause of death in the human population for decades [1]. Atherosclerosis of the peripheral and coronary arteries is one of the primary contributors to severe heart disease. Atherosclerosis is an immuno-inflammatory, chronic disease characterized by vascular remodeling and specific changes in the vessel walls. Endothelial cells, leukocytes, and smooth muscle cells participate in the development of atherosclerosis. Lipid deposition occurs on the walls of medium and large vessels during this process. Macrophages that have phagocyted a large amount of fats transform into foam cells, the majority of which die and undergo calcification. This process frequency increases with age. As a result of these changes, the lumen of the vessels narrows, leading to decreased blood flow, which may result in organ ischemia. In addition, the rupture of an atherosclerotic plaque may cause a thrombus formation, occluding the artery and causing sudden ischemia in the region previously supplied by the affected artery. Osteopontin (OPN) and osteoprotegrin (OPG) seem to be significantly involved in the atherosclerotic processes.

Osteopontin is a phosphorylated glycoprotein secreted by macrophages, vascular smooth muscle cells (VSMCs), and endothelial cells. OPN plays a key role in regulating biomineralization and skeletal remodeling [2,3]. Additionally, this glycoprotein is classified as a cytokine, because it transmits signals through integrin and CD44 receptors. It also participates in the migration and chemotaxis of macrophage and endothelial cells. As a cytokine, OPN is highly upregulated in both acute and chronic inflammatory conditions [4]. Its role in autoimmune diseases, liver diseases, and cancer is still not fully understood [5].

Osteoprotegerin, also known as osteoclast inhibitory factor (OCIT) or tumor necrosis factor receptor superfamily member 11B (TNFRSF11B), belongs to the tumor necrosis factor (TNF) receptor superfamily. OPG is a glycoprotein that inhibits skeleton resorption [6]. This process is possible because OPG prevents the receptor activator of the nuclear factor κB ligand (RANKL) becoming bound to the receptor activator of nuclear factor κB (RANK) [3,6]. The binding between RANKL and OPG can inhibit bone metastasis of tumors by suppressing bone resorption, preventing bone loss in adjuvant arthritis without affecting inflammation, and may serve as a pharmacological tool for osteoporotic and erosive bone disorders [6]. OPG-deficient mice present marked calcification of the aorta and renal arteries [6].

The American Heart Association (AHA) Atherosclerosis Scale is a histopathological tool used to distinguish atherosclerotic plaques based on morphological tissue criteria [7]. This scale directly correlates plaque type with the risk of cardiovascular events [7]. A modified AHA classification for imaging techniques further delineates eight types of plaques [7]. The project aims for a deeper understanding of the calcification process. Given that hydroxyapatite [Ca_10_(OH)_2_(PO_4_)_6_] is the primary component of calcium deposits formed in the arteries, and it is associated with extracellular proteins such as collagen and elastin, we decided to analyze, both quantitatively and qualitatively, the individual matrix proteins. So far, vascular analyses in the context of OPN/OPG have focused solely on type I collagen; however, we aim to expand the panel of analytes to include type III collagen (characteristic of the vascular system), elastin (a crucial protein for arterial vasodilatation), type IV collagen (which forms the structure of the basement membrane surrounding vascular muscle cells) [8], as well as type II collagen (characteristic of cartilage) [9].

## 2. Materials and Methods

During forensic autopsies performed at the Department of Forensic Medicine of Wroclaw Medical University, 101 aortic samples were collected. The study was approved by the Bioethics Committee of the Medical University of Wrocław (No 220/2010), and the use of the material for scientific purposes was authorized by the prosecutor on each occasion.

The samples were macroscopically graded as calcified or non-calcified (31 calcified/76 cases) and microscopically graded according to the American Heart Association scale of atherosclerosis. The fossilized lesion (*n* = 15) was isolated from the samples where calcium deposition was evident. The analyte content of the soft part of the artery was analyzed independently.

### 2.1. Measurement of Collagens Type I–IV and Elastin Content in Aortas’ Fragments by ELISA

The samples, of about 75 mm^3^, were homogenized in an extraction buffer (10 mM Tris, 5 mM EDTA, 0.2 M NaCl, pH 7.5) using a FastPrep-24^®^ homogenizer (MP Biomedicals, Santa Ana, CA, USA) in relation to 100 mg of tissue to 1 mL of buffer. The Maxisorp plates (Nunc^®^, Sigma-Aldrich, Darmstadt, Germany) were coated for 24 h at 4 °C with the homogenates. Standard solutions of collagen type I (Millipore, Billerica, MA, USA), collagen type II (Novus Biologicals, Centennial, CO, USA), collagen type III (Millipore), collagen type IV (Millipore, Billerica, MA, USA), elastin (Sigma-Aldrich, Darmstadt, Germany) were tested for the reference. The plates were blocked with 10% skim milk in PBST (PBS —phosphate buffered saline, 0.1% Tween 20) overnight at 4 °C. Then the primary antibodies were applied: mouse monoclonal IgG1 anti-collagen type I (Novus Biologicals, Abingdon, UK), rabbit polyclonal anti-collagen type II antibodies (Novus Biologicals), mouse monoclonal anti-collagen type III (Sigma-Aldrich), mouse monoclonal anti-collagen type IV (Sigma-Aldrich), or rabbit policlonal anti-elastin (Santa Cruz, Santa Clara, CA, USA). After a 2 h incubation, a secondary monoclonal anti-mouse IgG-HRP (H + L) (Jacson ImmunoResearch, West Grove, PA, USA) or anti-rabbit IgG-HRP (H + L) (Jacson ImmunoResearch) was applied for 1.5 h. Then the colorimetric reaction was developed using o-phenylenediamine (Sigma-Aldrich) and absorbance measured at 450 nm on an EnSpire Multimode Plate Reader (Perkin Elmer, Waltham, MA, USA). The concentration of proteins in the homogenates was calculated from the standard curves obtained for individual standard solutions of collagens and elastin. The same methodology was used in experiments described by Kuzan et al. 2014 and Kuzan et al. 2021 [8,9].

### 2.2. Measurement of OPN and OPG

The methodology was analogous to the above-mentioned antigens. Mouse Monoclonal Anti-human Osteopontin (R&D Systems, Minneapolis, MN, USA), Mouse Monoclonal Anti-human Osteoprotegerin/TNFRSF11B Antibody (R&D Systems) were used. The standard protein solutions used to perform the standard curve were Recombinant Human Osteopontin/OPN (R&D Systems) and Recombinant Human Osteoprotegerin/TNFRSF11B (R&D Systems). Peroxidase-conjugated rabbit anti-mouse IgG (H + L) was used as a secondary antibody (Jackson Immuno Research).

### 2.3. Immunohistochemical Staining of OPN and OPG

The methodology is analogous to that described in Kuzan et al. 2021 and Kuzan et al. 2017 [8,9]. Briefly, slides were dewaxed with xylene, hydrated with ethanol solutions, the antigens were discovered with Proteinase K (Dako, Minneapolis, MN, USA), endogenous peroxidase was blocked with Dako Real Peroxidase-Blocking solution and non-specific sites with DakoProtein Block (Dako). Then, reactions with the primary antibodies were carried out using Monoclonal Anti-human Osteopontin (R&D Systems, MAB14332) and Monoclonal Anti-human Osteoprotegerin/TNFRSF11B Antibody (R&D Systems, MAB8051). Then, the LSAB2 System-HRP, DakoCytomation kit was used for the next steps of immunohistochemical reactions, followed by an enzymatic reaction with the DAB substrate (3,39-diaminobenzidine, Dako North America, Carpinteria, CA, USA). The basophilic structures by Delafield’s hamatoxylin (Hematoxylin solution according to Delafield, Sigma) were stained, rinsed in tap water, dehydrated and fixed with ethanol and xylene, and sealed in DPX.

### 2.4. Statistical Analysis

The following statistical tests were used: Spearman correlation, Wilcoxon test, Mann–Whitney test, Kruskal–Wallis test, and F and *p* tests (ANOVA). Statistical analysis was performed using the data analysis software system Statistica (version 13.3, StatSoft, TIBCO Software Inc., Palo Alto, CA, USA), assuming *p* < 0.05 as the significance threshold.

## 3. Results

Basic descriptive statistics for the tested samples are presented in Table 1. We compared and sought relationships between various parameters to determine the content of osteoprotegerin and osteopontin in the aortic samples. No correlations were found between the content of OPN/OPG and gender/age.

The primary hypothesis was that the OPN/OPG content is significantly related to the degree of the atherosclerosis development. However, the used statistical tests did not confirm this hypothesis. Graphically, this result is presented in Figure 1, showing that there are no substantial differences between the groups in the content of the analyzed proteins.

However, a number of statistically significant correlations were observed among the tested proteins and other proteins of the extracellular matrix, especially between the OPG/OPN content and type I collagen, between the OPG/OPN content and type II collagen, between the OPN content and type III and IV collagen, and between the OPG/OPN content and elastin. All these correlations are positive and moderately strong. The specific parameters (r and *p* values) are listed in Table 2. These correlations are depicted graphically in Figure 2 and Figure 3.

Since the hypothesis assumed that OPN and OPG had an effect on tissue mineralization, it was expected that there would be more of these glycoproteins in the samples that were macroscopically assessed as calcified. The results of the comparison performed with the Mann–Whitney test are presented in Table 3 and graphically in Figure 4. The obtained results show there is no statistically significant difference in the OPG content between the two types of arteries, while the OPN content is statistically higher in calcified tissues than in uncalcified tissues.

As mentioned in the Materials and Methods, a calcium deposit was isolated from part of the aorta, the calcium plaque, and the soft part of the artery from the same patient. The samples were analyzed independently by means of paired-Wilcoxon tests and the results are presented in Table 4. Based on these data, we conclude that the OPG content is significantly higher in the artery. On the other hand, the OPN content seems to be lower in the artery and higher in the calcium deposit, although a statistical significance was not fully achieved (*p* = 0.073).

Immunohistochemical reactions with anti-OPN and anti-OPG antibodies showed that both proteins are either invisible or seen in trace amounts in tissues not involved in atherosclerosis (Figure 5). There is a relatively low expression of OPN content in regions of the artery intima that are at an advanced stage of atherosclerosis but not involved in calcification (regions of foam cells and others surrounding the calcium deposits), as shown in Figure 6A–C. OPN is mostly concentrated in regions of calcification, including isolated calcium deposits (Figure 6A–C). OPG, on the other hand, has quite a different localization—it tends to concentrate in the media of atherosclerotic lesioned aortas (Figure 6D,F). However, if the OPG concentrates in atherosclerotic plaques, the protein predominantly appear in the plaques’ fatty core region.

## 4. Discussion

The extracellular matrix (ECM) is essential for vascular function because it provides support for the vascular endothelium, media myocytes, and other vascular cells. ECM plays the primary role in maintaining mechanical or viscoelastic properties of the tissue. Three main components of the vascular ECM responsible for this are: elastic fibers, fibrillar collagens, and large aggregating proteoglycans [10]. However, numerous smaller proteins in the matrix fulfill a regulatory role, such as thrombospondin, tenascin, alkaline phosphatase (ALP), bone morphogenetic protein (BMP), OPN, OPG, calpain-1, runt-related transcription factor (Runx-2), and many others [11,12]. The present study focuses on the interaction between OPN/OPG and different collagen types (I–IV) and elastin, which are major extracellular matrix proteins.

Osteoprotegerin and osteopontin are known as vascular calcification inhibitors in the atherosclerotic and inflammatory processes. It has been postulated that the determination of these proteins in plasma may be of diagnostic value. More than a decade ago, it was observed that the concentration of OPG and OPN in serum was associated with arterial stiffness and the presence and severity of coronary artery disease [13]. Recent articles have proved that high concentrations of OPG and OPN are associated with atherosclerosis-based conditions, such as acute myocardial infarctions [14], peripheral arterial disease (PAD), or other adverse cardiovascular events [15]. The OPN and OPG concentration is higher in symptomatic or unstable atherosclerosis compared to the concentration level in cases of asymptomatic and stable atherosclerosis. Therefore, the concentration level can serve as an excellent diagnostic marker for estimating the risk of plaque rupture and the occurrence of a heart attack or stroke [3].

Lin et al. suggested that concentrations of OPG and OPN are associated with increased mortality rates in patients with coronary artery disease [16]. However, the predictive value of both OPN and OPG in assessing clinical outcomes in patients with peripheral artery disease is unclear. Simultaneously, it was indicated that elevated levels of OPG or OPN may predict higher mortality in patients with peripheral artery disease [16]. The measurements of OPG and OPN levels should not be used as the sole markers for identifying endothelial dysfunction, they could serve as additional clinical biomarkers for endothelial dysfunction and prognostic parameters for predicting vascular events [17]. Furthermore, the blood concentrations of OPG and OPN could be utilized in conjunction with, and interpretation of, MRI images [18]. It is important to note that both OPN and OPG are not routinely included as laboratory parameters in cardiology wards.

In light of the evidence that testing the serum concentration of OPN and OPG may have diagnostic significance, it seems obvious that in this project, in addition to arterial samples, it would be worthwhile to also analyze the levels of these analytes in the blood of the same patients. The results would then be obtained for paired samples, enabling an analysis of how much the antigen concentration in the tissue translates into the amount circulating in the vascular system. This approach is preferentially used by authors (for example in Kuzan et al. 2021 [19]). Unfortunately, in a project based on tissues from deceased persons, it is not possible to perform such an analysis because it is not possible to collect a reliable blood sample from the deceased. It is admitted that this is one of the limitations of this project. Another important limitation is that for legal reasons, we do not have access to the medical history of tissue donors—this information would also enrich the project.

What distinguishes our work from others is a large sample database, drawn from over 100 different patients. Contrary to serum samples, obtaining aortic samples is challenging, which renders them valuable material for research. Simultaneously, immunohistochemical staining of OPN and OPG in tissue samples differs from serum tests and cannot be considered a search for a diagnostic marker. Our study does not aspire to be an applicative study; rather, it belongs to basic research aimed at investigating the pathomechanism of cardiovascular diseases. Unfortunately, there are few publications describing tissue analyses that could be referred to.

Immunohistochemistry performed by Golledge et al. demonstrated that OPG and OPN were mainly distributed around calcification areas as well as inflammatory cells and VSMCs [3]. Wolak et al. also report that a high OPN concentration in plaque areas correlates with plaque calcification [20]. In general, OPN is detected in atherosclerotic plaques and the areas of arterial ischemia. OPN function as a cytokine is essential for macrophage infiltration and post-ischemic neovascularization [4].

Our work is also notable for comparison with the study by Strobescu-Ciobanu et al., who investigated carotid artery tissues. They described strong expression of OPG in stable calcified plaques, while OPN expression was minimal. OPN was primarily found in the periphery of the calcification area, within foamy macrophages, fibroblasts, and plaque inflammatory cells [21]. However, our findings do not align this observation. We discovered that OPN is not only present in the periphery of the calcified area but also in its center. OPN expression seems to be strong in calcified plaques, which appears to be confirmed by our quantitative ELISA tests. Nevertheless, similar to our study, the aforementioned authors did not establish a relationship between OPN/OPG and the type/stage of plaque according to the AHA [21].

Vascular calcification is an age-related phenomenon that occurs in various arteries, ranging from small coronary arteries to the aorta. The more intense the calcification, the more rigid and prone to cracking the tissue becomes [22]. We can confirm the previous reports that OPN is not detected in significant amounts in the walls of healthy arteries but is found in significant amounts in the calcified walls of human vessels [23]. In the work of Giachelli et al., it has been proven that a high phosphate concentration causes the loss of smooth muscle cell line markers and the simultaneous increase of osteochondrogenic markers, including OPN [23,24]. In vitro studies on cell cultures have shown that OPN functions by directly binding to the surface of hydroxyapatite crystals, suggesting that OPN may bind to the growing crystal surfaces and potentially inhibit further crystal growth [24].

The lack of correlation between OPN and OPG is another interesting observation in our research. While both proteins play an anti-calcifying function, one might expect them to correlate with each other. Similarly, Higgins et al. also found no such correlation between OPN and OPG while analyzing the carotid artery [22]. Thus, it can be concluded that although the proteins in question play a common function, not only do they have different locations but also their concentrations do not uniformly increase, and their expression is regulated differently.

It is reported that factors stimulating OPG expression include vitamin D, transforming growth factor β, bone morphogenetic protein-2, and estrogen [25]. OPN synthesis is also stimulated by estrogens and inhibited by testosterone [4]. Therefore, it can be suspected that OPG and the OPG concentration will be higher in women. However, our research did not confirm this statement, as we did not observe any relationship between sex and the level of OPG and OPN in the tissue. This may be attributed to the small size of our study group and/or the fact that the women included were predominantly postmenopausal (the average age of the patients was 55 years) [4].

As regards the age influence on OPG expression, most of the epidemiological studies have demonstrated a positive correlation between age and OPG content [25,26,27]. However, we have noticed that the correlation applies only to the OPG concentration in serum and not to that in arterial tissue. Some researchers, such as Nawaz et al., did not find a relationship between serum OPG and age [28]. The situation is similar when it comes to OPN; there are some reports which confirm the lack of relationship between serum OPN and age [29,30], while others show a positive correlation [31]. Our study did not confirm the trend of OPN/OPG concentration increasing with age, although this result may be due to the fact that our study group was perhaps too homogeneous in terms of age.

What distinguishes our work from other studies in the field is the fact that we marked other extracellular matrix proteins in detail. Crystallographic and other methods have proven that OPN interacts with the extracellular matrix by binding to collagens I, II, III, IV, and V. The interaction is Ca2+-dependent and requires non-denatured triple helical collagen [32,33]. But it is still an open question if there is a quantitative relationship between collagen and OPN.

Collagen and OPG relationship in the arteries have been confirmed by a few researchers. One of them was Gajewska et al., who demonstrated that OPG correlates with C-telopeptide of type I collagen (s-CTX) in the serum of healthy children and adolescents [34]. The researchers focused on the OPG relationship to bone mineralization during adolescence rather than pathological processes such as atherosclerosis. Meng et al. studied OPN and collagen content in the aorta, specifically in cases of ascending aortic aneurysms (ATAA), and observed that both collagen and OPN increased in ATAA [35]. In patients with hypertensive heart disease, heart failure, and dilated cardiomyopathy, it has been proven that increased myocardial OPN content strongly correlates with collagen type I expression and insoluble collagen [36,37]. However, it is important to note that we examined aortic tissue but not myocardial tissue, so it is difficult to compare our studies directly (although we observed an OPN/type I collagen correlation).

The reviews report that osteonectin expression is correlated with fibrillar collagen expression [32]. These claims are supported by the following evidence: (1) OPN-null mice demonstrate reduced collagen deposition versus that in wild-type mice [38]; (2) targeting OPN expression by siRNA in fibroblasts from individuals with scleroderma reduces collagen type I expression in vitro [32]; and (3) delivery of exogenous OPN on dermal fibroblasts has a pro-fibrotic effect [39]. We were the first to provide direct evidence that the content of fibrillar collagen (types I and II) in aortic tissue increases with an increasing OPN level. It can be suspected that OPN and OPG, by stimulating collagen expression, have a compensatory effect on vascular endothelial damage. The compensatory process aims to cover the plaque with a layer of matrix that will protect the fatty and/or thrombogenic core of the plaque against rupture and release of content into the bloodstream [40].

The analysis of the results allows us to plan future research and experiments, such as testing to define the correlation between parameters that determine collagen synthesis or breakdown, e.g., pyridinoline (Pyr), deoxypyridinoline (DPyr), N-terminal propeptide of type I procollagen (PINP, P1NP), C-terminal propeptide of type I procollagen. (PICP, P1CP), and LOX (lysyl oxidase). This research will provide a broader understanding and enable conclusions regarding mechanisms. It would be worthwhile to conduct similar studies on a larger group of patients spanning a wider age range, especially to including women in the premenopausal age (so that more reliable conclusions about the relationship between OPN and OPG in the arteries with age and sex could be made).

## 5. Conclusions

Arterial calcification is a vital common issue. Upon dissecting the aorta, macroscopic aortic calcification was visible in more than half of the patients. We have confirmed that there is a relatively large amount of OPN in the calcium deposit structure. Conversely, OPG is primarily found on the periphery of the calcifying areas and even in the parts of the artery that have not yet been atherosclerotic, such as the media. The molecular relationship between the glycoproteins and the expression of other extracellular matrix proteins is not clear. However, we report that OPN and OPG, although not correlated with each other, exhibit a correlated content in tissue with fibrillar collagens, such as collagen types I, II, and III, as well as type IV collagen and elastin.

## Figures and Tables

**Figure 1 biomedicines-12-00847-f001:**
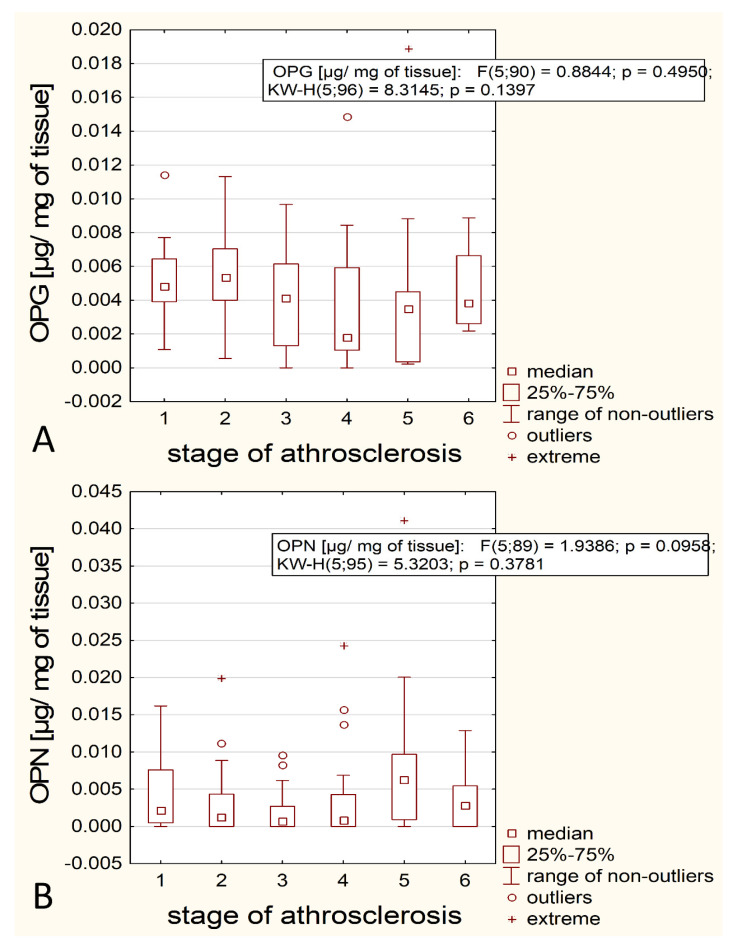
Dependence of OPG and OPN on the degree of atherosclerosis (**A**,**B**).

**Figure 2 biomedicines-12-00847-f002:**
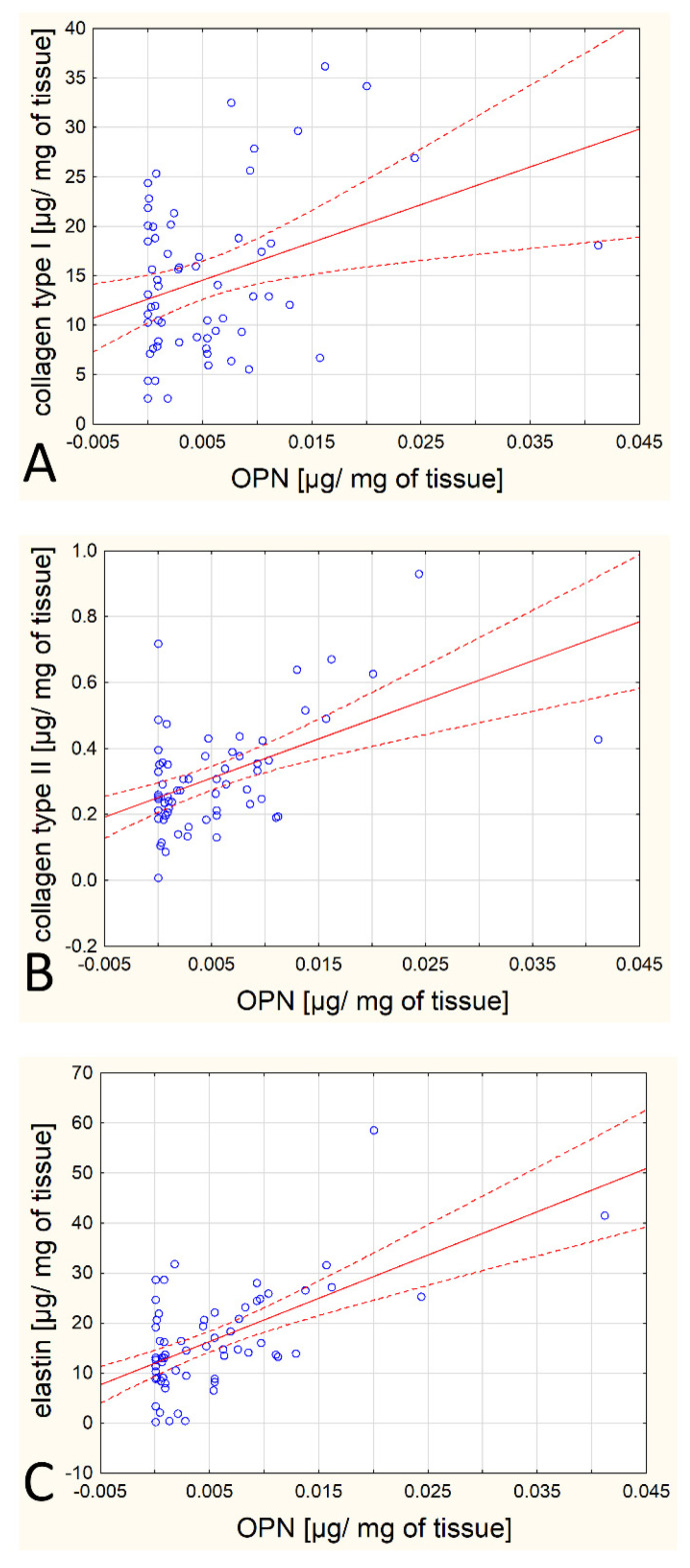
Scatterplot for OPN and other analytes for which a statistically significant correlation was obtained (**A**–**C**).

**Figure 3 biomedicines-12-00847-f003:**
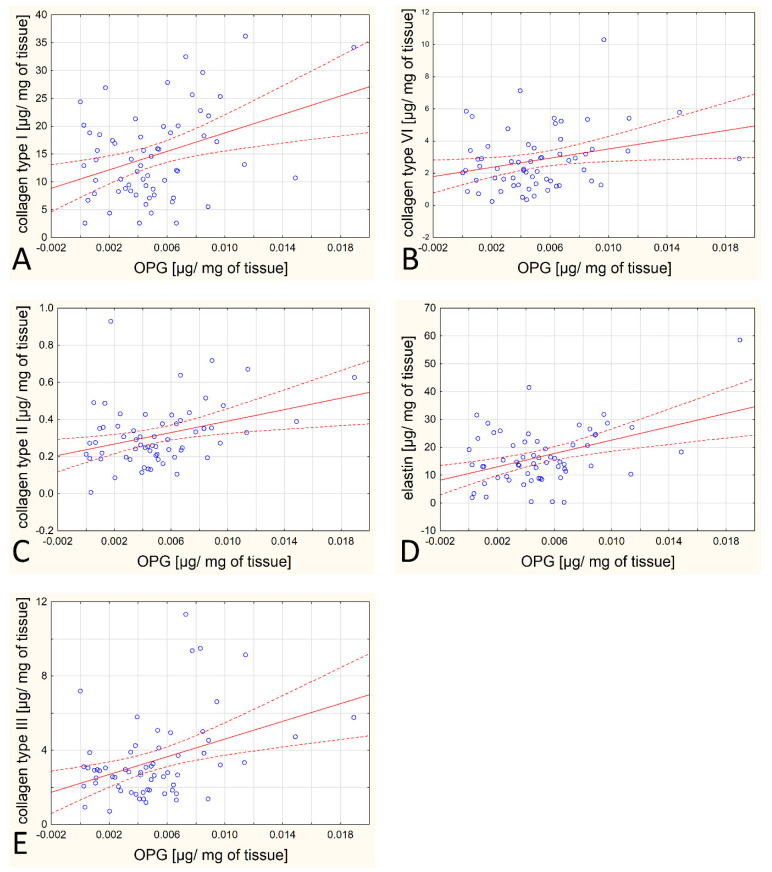
Scatterplot for OPG and other analytes for which a statistically significant correlation was obtained (**A**–**E**).

**Figure 4 biomedicines-12-00847-f004:**
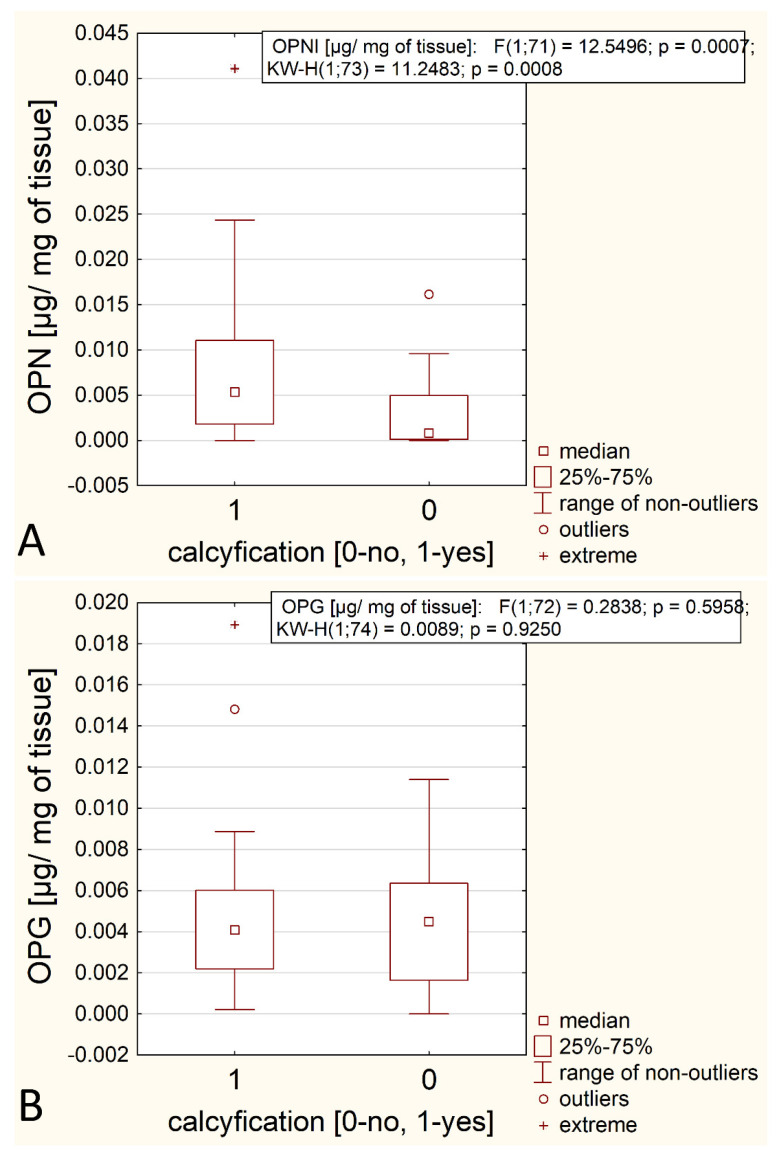
Comparison of the OPN and OPG contents in calcified and uncalcified samples (**A**,**B**).

**Figure 5 biomedicines-12-00847-f005:**
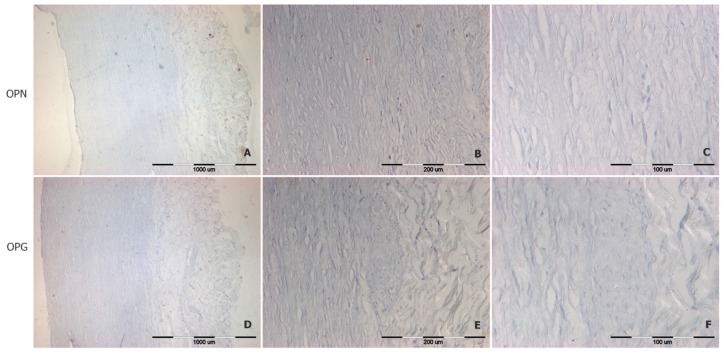
Representative immunodetection images of OPN (panels **A**–**C**) and OPG (panels **D**–**F**) in the aortic wall of a person with very low-grade atherosclerosis, without signs of calcification. 40× magnification (panels **A**,**D**), 200× (panels **B**,**E**), 400× magnification (panels **C**,**F**).

**Figure 6 biomedicines-12-00847-f006:**
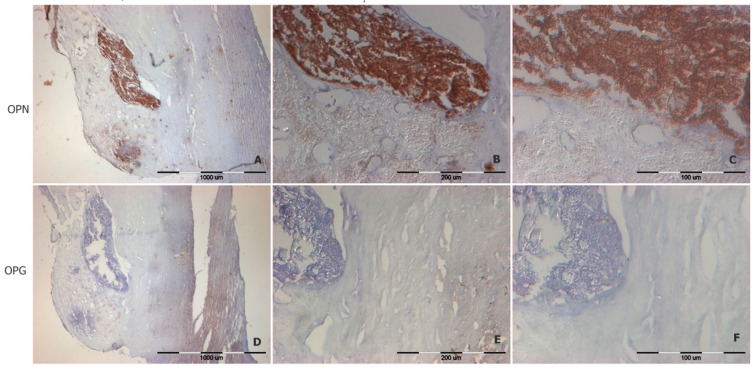
Representative immunodetection images of OPN (panels **A**–**C**) and OPG (panels **D**–**F**) in the aortic wall of a person with advanced atherosclerosis, with a visible calcium deposit. Magnification 40× (panels **A**,**D**), 200× (panel **B**,**E**), magnification 400× (panels **C**,**F**).

**Table 1 biomedicines-12-00847-t001:** Basic descriptive statistics for the tested samples.

	*N*	Mean	Minimum	Maximum	Standard Dev.
Gender [0—women; 1—men]	100	0.710	0.000	1.000	0.456
age	101	55.396	18.000	90.000	15.868
Calcification [0—no, 1—yes]	76	0.408	0.000	1.000	0.495
OPG [µg/mg of tissue]	101	0.004	0.000	0.019	0.003
OPN [µg/mg of tissue]	100	0.004	0.000	0.041	0.006

**Table 2 biomedicines-12-00847-t002:** Abbreviated correlation matrix for OPN, OPG, and other analyzed arterial extracellular matrix proteins and patient age. Correlations that achieved statistical significance (*p* < 0.05) are marked in red.

	Collagen Type I [µg/mg of Tissue]	Collagen Type II [µg/mg of Tissue]	Collagen Type III [µg/mg of Tissue]	Collagen Type IV [µg/mg of Tissue]	Elastin [µg/mg of Tissue]	Age [Years]	OPG [µg/mg of Tissue]
OPG [µg/mg of tissue]	0.3696	0.3378	0.3911	0.2746	0.4219	−0.0817	-
	* p * = 0.003	* p * = 0.007	* p * = 0.002	* p * = 0.031	* p * = 0.001	*p* = 0.528	-
OPN [µg/mg of tissue]	0.3401	0.5172	0.0974	0.0851	0.6073	0.1072	0.1567
	* p * = 0.007	* p * = 0.000	*p* = 0.452	*p* = 0.511	* p * = 0.000	*p* = 0.407	*p* = 0.224

**Table 3 biomedicines-12-00847-t003:** Comparison of the OPN and OPG contents in calcified and uncalcified samples. P—percentile.

	Uncalcified				Calcified				
	Mean	SD	Median	25–75 P	Mean	SD	Median	25–75 P	*p*-Value
OPG	0.005	0.003	0.005	0.002–0.006	0.005	0.004	0.004	0.002–0.006	0.791
OPN	0.003	0.006	0.001	0.000–0.005	0.008	0.009	0.005	0.002–0.011	0.001

**Table 4 biomedicines-12-00847-t004:** Comparison of the OPN and OPG contents in arteries and calcium deposits isolated from the same artery. P—percentile.

		Mean	SD	Median	25–75 P	*p*-Value
OPG [µg/mg of tissue]	Artery	0.004	0.005	0.002	0.001–0.006	0.002
	Calcium deposit	0.001	0.004	0	0.000–0.001	
OPN [µg/mg of tissue]	Artery	0.006	0.007	0.004	0.000–0.011	0.073
	Calcium deposit	0.017	0.020	0.011	0.006–0.020	

## Data Availability

Raw data available from the corresponding author upon reasonable written request.

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
