# Peer review of "Relationships between Osteopontin, Osteoprotegerin, and Other Extracellular Matrix Proteins in Calcifying Arteries"

_biomedicines, 2024, doi:10.3390/biomedicines12040847_

Round 1

Reviewer 1 Report

Comments and Suggestions for Authors

The study on arterial calcification might be of interest. The rough preparation in paper might be improved to read easily.

1.      The comparison of OPN and OPG between blood and tissue would be more of interest. Could there be any correlation between blood OPN and OPG and tissue ones in your study? If it is supposed, the discussion might be more included in the text. Are there any other molecules of having some correlation between blood and tissue levels?

2.      Arterial calcification is partly related to skeletal and bone aging. Are there any more discussion from your findings? If possible, the in-depth suggestion would be helpful for the future work.

3.      Methods; Macroscopically graded calcification according to the American Heart Association scale of Atherosclerosis could be briefly introduced in some sentences (row 84-85).

4.      Methods; the methodology could be briefly described (row 89).

5.      Methods; in the respective assay, the measuring performances (coefficient of variation, detection level, reproducibility, needed sample volume etc.) could be added.

6.      Methods; Spearman correlation, Wilcoxon test, Mann-113 Whitney test, Kruskal-Wallis test, F and p test (ANOVA); the cases to use these tests (e.g. parametric values) were described. What was P test?

7.      Abstract; ELISA could be fully spelled out in the first appearance (row 21).

8.      Results; significant digits could be rechecked in the expression of measured values and statistical results while considering precision and accuracy.

9.      Table 2 and 4; some red texts were found (why?).

10.   Table 1; maksimum/maximum (which seemed to be better).

11.   Row 23; correlations between; space appeared to be wide, so check the space. There were many parts of such wide spaces (Row 46, 49, 57, 58, 61, 68, …). Check the overall text.

12.   Row 29; OPN/ elastin; space could be deleted before elastin.

13.   Row 115; Statistica (version 13.3, StatSoft); city or country name for the software product could be added.

14.   Row 101; ..et al 2017.[7,8].; ‘.’ could be deleted after 2017.

15.   Row 101; Briefly:; ‘:’ could be changed to ‘,’.

16.   Row 121-122; OPN / OPG; space could be deleted before and after /.

17.   Row 122; the expression of ‘correlation’ and ‘were’ was mismatched (singular or plural?).

18.   Row 208; marker., Our…; ‘.,’ was a mistaken part.

19.   Row 235; osteopontin could be abbreviated like the prior parts.

20.   Row 305; osteoprotegerin could be abbreviated like the prior parts.

Comments on the Quality of English Language

Extensive editing of English language required.

Reviewer 2 Report

Comments and Suggestions for Authors

1. The results of this manuscript could not support the conclusion due to insufficient evidence.

2. The quality of immunodetection for osteopontin, osteoprotegerin, and other extracellular matrix proteins was low which cannot support the hypothesis of this study.

3. The manuscript is more descriptive without mechanistic insights.

4. The results of immunostaining should be quantified.

5. The style of references should be revised.

Comments on the Quality of English Language

Minor editing of English language required

Reviewer 3 Report

Comments and Suggestions for Authors

The authors have submitted a research article of studying a relationship between expression regulation of both osteopontin (OPN) and osteoprotegerin (OPG) and fibrillar collagens and elastin in order to verify the mechanism of arterial calcification in atherosclerosis. It has been well recognized that during the process of atherosclerosis, vessels are remodeled, resulting in calcification due to inflammation. Generally, OPN and OPG function to the same direction, i.e. inhibition of calcification. In this regard, it has been hypothesized that the expression regulation of both OPN and OPG is similar to each other, but their study process found no statistically positive associations of the two glycoproteins, which might have a potential to be an important, unrevealed knowledge regarding suppression of calcification by the expression of both OPN and OPG. This issue is of interest, but impact of their article is moderate. My overall concern with the article describing the available data regarding a possible, beneficial availability of both OPN and OPG expression which might predict risk of atherosclerosis in patients is that information provided may offer something substantial that helps advance our understanding of effective management of atherosclerosis which draws novel class of effective medicinal compounds available in clinic.

The therapeutic prediction based on the present data regarding glycoproteins of both OPN and OPG is definitely the primary goal of this submission, so that readers are likely to know how to predict the effective diagnosis on atherosclerosis, based on the information regarding the data like this study. The authors should discuss this issue in their revision.

Round 2

Reviewer 1 Report

Comments and Suggestions for Authors

The paper was improved.

1.      Even though comparing tissue and blood samples from a single patient was impossible in this study, the future advance with the papers (e.g., DOI: 10.3390/biom11040557, the other is in review in Thyroid Research) could be more discussed in the discussion or as the limitation.

2.      In statistics, what was P test? The test could be explained by some papers.

3.      The overall writing, including the use of spaces and abbreviations) should be rechecked.

Comments on the Quality of English Language

The overall writing, including the use of spaces and abbreviations) should be rechecked.

Author Response

1. Even though comparing tissue and blood samples from a single patient was impossible in this study, the future advance with the papers (e.g., DOI: 10.3390/biom11040557, the other is in review in Thyroid Research) could be more discussed in the discussion or as the limitation.

We have added the paragraph to the discussion section as per your suggestion:

“In light of the evidence that testing the serum concentration of OPN and OPG may have diagnostic significance, it seems obvious that in this project, in addition to arterial samples, it would be worthwhile to also analyze the levels of these analytes in the blood of the same patients. Results would then be obtained for paired samples, enabling analysis of how much the antigen concentration in the tissue translates into the amount circulating in the vascular system. This approach is preferentially used by authors (for example in Kuzan et al 2021 [19]). Unfortunately, in a project based on tissues from deceased persons, it is not possible to perform such an analysis because it is not possible to collect a reliable blood sample from the deceased. It is admitted that this is one of the limitations of this project. Another important limitation is that for legal reasons, we do not have access to the medical history of tissue donors - this information would also enrich the project.’

We hope this significantly improves the article.

  1. In statistics, what was P test? The test could be explained by some papers.

The p-test is one of the methods used in analysis of variance (ANOVA). It calculates an F-statistic and associated p-value to determine whether there are statistically significant differences between the group means.

In the text, we stated, 'The following statistical tests were used: Spearman correlation, Wilcoxon test, Mann-Whitney test, Kruskal-Wallis test, F-test, and p-test (ANOVA).' Both the F-test and the p-test are components of ANOVA.

  1. The overall writing, including the use of spaces and abbreviations) should be rechecked.

We have once again revised and corrected all spacing and abbreviations. Additionally, we have rechecked the English language for accuracy.

Reviewer 2 Report

Comments and Suggestions for Authors

The authors cannot provide convincing data for their revision due to the limitation of experimental equipment. 

Author Response

Thank you for re-examining our manuscript. Unfortunately, we are not sure what the Reviewer means by "experimental equipment". If it is about laboratory equipment, we admit that we are surprised that the reviewer discredits our equipment, because we have already published a lot of manuscripts in which we present results obtained with exactly the same equipment - mainly the ELISA reader and the Olympus BX microscope, and no previous reviewer has discredited the quality image. For example, some works:

  1. Kuzan A, Królewicz E, Nowakowska K, Stach K, Kaliszewski K, Domosławski P, Kotyra Ł, Gamian A, Kustrzeba-Wójcicka I. Contribution of Glycation and Oxidative Stress to Thyroid Gland Pathology-A Pilot Study. 2021 Apr 10;11(4):557. doi: 10.3390/biom11040557. PMID: 33920190; PMCID: PMC8069218.
  2. Kuzan A, Wujczyk M, Wiglusz RJ. The Study of the Aorta Metallomics in the Context of Atherosclerosis. 2021 Jun 25;11(7):946. doi: 10.3390/biom11070946. PMID: 34202347; PMCID: PMC8301911.
  3. Kuzan A, Wisniewski J, Maksymowicz K, Kobielarz M, Gamian A, Chwilkowska A. Relationship between calcification, atherosclerosis and matrix proteins in the human aorta. Folia Histochem Cytobiol. 2021;59(1):8-21. doi: 10.5603/FHC.a2021.0002. Epub 2021 Feb 9. PMID: 33560515.
  4. Kuzan A, Chwiłkowska A, Pezowicz C, Witkiewicz W, Gamian A, Maksymowicz K, Kobielarz M. The content of collagen type II in human arteries is correlated with the stage of atherosclerosis and calcification foci. Cardiovasc Pathol. 2017 May-Jun;28:21-27. doi: 10.1016/j.carpath.2017.02.003. Epub 2017 Feb 14. PMID: 28284062.
  5. Kuzan A, Smulczyńska-Demel A, Chwiłkowska A, Saczko J, Frydrychowski A, Dominiak M. An Estimation of the Biological Properties of Fish Collagen in an Experimental In Vitro Study. Adv Clin Exp Med. 2015 May-Jun;24(3):385-92. doi: 10.17219/acem/31704. PMID: 26467125.

We will also add that the other two reviewers assessing this manuscript also did not comment on the quality of the photos or the laboratory equipment in general.

We would like to add that we hope that, despite methodological limitations, the reviewer will find our work worth publishing. We would like to emphasize that due to the relatively large amount of very valuable material examined, such as arterial sections, and the relatively wide panel of antigens that we analyzed, our work is unique and valuable for basic research.

Reviewer 3 Report

Comments and Suggestions for Authors

The authors have done a good job responding to reviewer comments and concerns in their revision. I believe the manuscript is significantly improved as a result. Now I recommend that this revised version of the manuscript can be accepted for publication in Biomedicines.

Author Response

We are deeply appreciative of all the advice provided on how to enhance the article. Furthermore, we are delighted by the acceptance for this version of publication in Biomedicines by reviewer.

Round 3

Reviewer 2 Report

Comments and Suggestions for Authors

No further comment.